# Dynamic Properties of Pretreated Rubberized Concrete under Incremental Loading

**DOI:** 10.3390/ma14092183

**Published:** 2021-04-24

**Authors:** Aijiu Chen, Xiaoyan Han, Zhihao Wang, Tengteng Guo

**Affiliations:** 1School of Civil Engineering and Communication, North China University of Water Resources and Electric Power, Zhenzhou 450045, China; caj@ncwu.edu.cn (A.C.); wangzhihao@ncwu.edu.cn (Z.W.); guotth@ncwu.edu.cn (T.G.); 2School of Water Conservancy, North China University of Water Resources and Electric Power, Zhenzhou 450045, China

**Keywords:** pretreated rubberized concrete, stress level, dynamic properties, natural frequency, flexural dynamic stiffness, damping ratio

## Abstract

Recycling scrap tyres as alternative aggregates of concrete is an innovative option. To clarify the dynamic properties of the pretreated rubberized concrete with some cumulative damage, the natural frequency, flexural dynamic stiffness, and damping ratio of the specimens under incremental stress level were investigated in this paper. The results indicated that the pretreatment of rubber particles improved the strength, ductility, and crack resistance of the rubberized concrete. The reduction of the flexural dynamic stiffness was clarified with the increase of concrete stress level. The addition of the pretreated rubber particles enhanced the concrete energy dissipation capacity during the destruction, and the specimen dissipated more energy with the increase of rubber content before its failure.

## 1. Introduction

In recent years, the continuous development of new raw materials has enriched the materials in the field of construction [1,2,3,4], among which the treatment and recovery of solid waste are of great significance for environmental protection, such as the utilization of recycled tyre polymer fiber as concrete reinforced materials by Chen et al. [1], the incorporation of construction and demolition wastes as recycled aggregates by Rodríguez et al. [2], and the utilization of waste tyre rubber as concrete material by Huang et al. [3] and Gupta et al. [4]. The scrap tyre has become one of the most problematic solid wastes throughout the world, and many researchers have proved that scrap tyres are sources of valuable raw materials [5,6,7]. The application of rubbers recovered from waste tyres as concrete materials is beneficial to the environment and the sustainable development of society [8,9,10].

The rubberized concrete (RUC) with the appropriate amount and size of rubber particles featured better durability [11], thermal and sound insulation properties [12,13] as well as ductility [14,15] than those of normal concrete (NC). Aslani et al. [16,17,18] studied the properties of high-performance self-compacting concrete produced with rubber and achieved remarkable results. However, the concrete strength reduced significantly by replacing aggregates with rubber particles [19,20,21]. Compared with NC, the compressive strength, splitting tensile strength, and elastic modulus of the concrete where 20% fine aggregate was replaced with rubber particles were reduced by 30% at least, and with the concrete where 20% coarse aggregate was replaced, the reduction was larger [20,22,23,24,25,26]. Numerous methods to increase the properties of RUC and rubberized mortars have been proved feasible, such as pretreatment of rubber particles, application of fibers, etc. [18,27,28,29,30,31].

In recent years, researchers have paid more and more attention to concrete’s dynamic properties [32,33]. The previous literature was mainly about the properties of unpretreated rubberized concrete without damage. As a green potential concrete material, the dynamic properties of RUC have become an important research topic in recent years [34,35,36,37,38]. Chi et al. [39] and Xue and Shinozuka [40] researched the energy-dissipation capability of rubberized concrete, and showed that the concrete curvature ductility and damping ratio were enhanced significantly by replacing aggregate with rubber particles. Xue and Shinozuka [40] reported that the damping ratio of the RUC mixed with 15% rubber particles with a maximum size of 6 mm increased by 62% compared with that of NC. The damping ratio of the concrete whose coarse or fine aggregate was replaced with rubber particles was enhanced significantly [26,37,41]. The particle size, content and mixing method of rubber particles had significant effects on the concrete’s dynamic properties [26,42,43]. As Gurunandan et al. [26] reported, the damping ratio of the RUC with 7.5% and 22.5% fine aggregate replaced with rubber particles increased by 33.0% and 77.1% compared with that of NC at 56-day age. Also, Zheng et al. [42] reported that when using rubber particles instead of 15–45% coarse aggregate, the damping ratio of the RUC increased by 19.2–75.3% and 28.6–144.0% when the size of the rubber particles was 2.62 mm and 15.00–40.00 mm, respectively.

According to the above studies, rubber particles replacing coarse aggregate increased the concrete damping capacity more notably, but the strength reduced significantly. The concrete with fine aggregate replaced with rubber particles was relatively better. Muñoz-Sánchez et al. [28] reported that the surface hydrophilicity, absorption, and roughness of rubber particles increased after acid or alkaline treatments. Mohammadi et al. [27], Si et al. [29], Guo et al. [44], and Rivas-Vázquez et al. [45] showed that pretreated rubberized concrete (PRC) could maintain good workability, mechanical properties and enhance durability with reduced environmental impacts. Compared with NC, the compressive strength of the PRC reduced by only 14% (28 days), after replacing 10% fine aggregate with rubber particles pretreated with alkaline activation [46]. Therefore, a further study on dynamic properties of concrete with fine aggregate replaced with pretreated rubber particles is necessary.

Numerous mentioned papers have drawn important conclusions about the damping capacity of RUC through different methods, and promote the development of research on the dynamic capacity of concrete [40,42,43]. As reported by Xue and Shinozuka [40], Zheng et al. [42], and Najim and Hall [43], the change trend of the damping ratio with the increase of rubber content was almost the same, but the test values of damping ratio and natural frequency showed a significant difference due to the different test methods and specimens of different size and shape. The frequencies of concrete specimens were hundreds to thousands Hz, and the energy dissipation capability was significantly affected by the incompletely fixed ends and other vibration factors. Just as Strukar et al. [14] said, the inconsistent effects of recycled rubber in concrete on the static and dynamic properties of concrete materials and structures indicated that there was vital need for further research.

The reported publications mainly studied the properties of original rubberized concrete with undamaged specimens. The dynamic properties of concrete with some damage directly affect the anti-vibration capacity of the structures. It is necessary to study the dynamic properties of PRC during its destruction. In this test, specimens of uniform size were used, and the dynamic properties of the PRC, including natural frequency, flexural dynamic stiffness and damping ratio under incremental loading, were investigated using free vibration tests on them. The raw materials and the test methods were stated first, and then a brief research on the concrete slump and mechanical properties was made. Finally, the concrete dynamic properties were investigated.

## 2. Materials and Methods

### 2.1. Materials

The cement used in this test was the 42.5 Ordinary Portland Cement produced by Tianrui Group Cement Co., Ltd., Zhengzhou, China. The chemical compositions and properties of the cement are listed in Table 1, the preparation of cement mortar specimens and the test of their compressive and flexural strength are show in Figure 1.

The coarse and fine aggregates were natural crushed stone and river sand. Their properties are listed in Table 1. The apparent density of the rubber particles with the size of 1–2 mm is 1250 kg/m^3^, and the ash content, fiber content and moisture content are 2.7%, 0.5%, and 0.6%, respectively. The distributions of the rubber and aggregate particle size are shown in Figure 2, and the test of coarse aggregate crushed index and the distributions of the rubber and aggregate particle size are shown in Figure 3.

The pretreated solution of rubber particles was sodium hydroxide solution with a mass concentration of 5%. The rubber particles were washed with water three times and dried, then soaked in the solution for 30 min. The mass ratio of rubber particles and the solution was 1/2, and each batch of the solution was used only once on the rubber particles. Finally, they were washed with water again, and dried.

### 2.2. Concrete Mixes and Specimen Preparation

The concrete mixtures were based on China Standards JGJ 55- contents of the rubber particles were 0%, 5%, 10%, 15%, and 20%. The dynamic 2011 [47]. The volumes of sand properties and frost resistance may decrease when the rubber content exceeds 20% based on the previous literature [35,48], hence a maximum value of 20% rubber particles was used in this test. The mix proportions of RUC and PRC are listed in Table 2.

A total of nine groups of specimens, including four groups of RUC, four groups of PRC, and a group of NC as reference specimens were prepared. Each group contained 17 concrete specimens, three for the test of dynamic elastic modulus and flexural strength, two for the test of dynamic properties, and three for each other test. The six specimens for the test of cube compressive strength and splitting tensile strength were of the size of 100 × 100 × 100 mm. Another six specimens for the test of axial compressive strength and static elastic modulus were of the size of 150 × 150 × 300 mm. Three specimens for the test of dynamic elastic modulus and flexural strength were in the size of 100 × 100 × 400 mm. The last two specimens for the test of dynamic properties were in the size of 100 × 100 × 1000 mm.

The structural form of the cantilever beam is simple, and the pre-loading damage is easy to operate [49,50]. The natural frequency of specimens can be calculated with the dynamic elastic modulus, the mass and the length–width ratio of the specimens [50]. Based on the normal concrete cantilever beam with the natural frequency of about 30 Hz, rubberized concrete specimens of small cantilever beams were used for the free vibration tests. The longitudinal reinforcement, with a diameter of 6 mm and a grade of HPB300, was 432 MPa at yield and 556 MPa at peak. The hooping was made with iron wire of a diameter of 4 mm. The thickness of the protective layer was 15 mm. Specimens of each group were produced at the same time, and the tests were performed at the end of 28 days curing period, keeping the specimens in a curing room within 20 ± 5 °C and over 95% for the relative humidity.

### 2.3. Test Methods

The properties of concrete including the slump, strength, static, and dynamic elastic modulus were tested based on China Standards GB/T50080-2016 [51], GB/T50081-2002 [52] and British Standards BS 1881-209-1990 [53], respectively. For each test, the average value of the three specimens was reported as the test value. The natural frequency, flexural dynamic stiffness and damping ratio were obtained through the free vibration test of the small cantilever beams under incremental loading (0, 0.8*M_r_*, 0.6*M_u_* and 0.8*M_u_*) with 0–0.8 of the concrete stress level ($\bar{\sigma }$), as shown in Figure 4.

Three concrete stress levels including 0.8*M_r_*/*M_u_*, 0.6 and 0.8 were designed, and the dynamic properties of concrete both before and after cracking were studied, where *M_r_* and *M_u_* were the flexural cracking and ultimate moment of the cantilever beam. As shown in Figure 4, the concrete stress level was prepared under a concentrated load with a hanging basket and a set of calibrated weights by using a standard graded loading procedure based on China Standards GB/T 50152-2012 [54]. Based on the assumptions and method of calculation in Standards GB50010-2010 [55], the stress of concrete during the destruction was determined by
(1)$$\sigma_{c}\;=\;E_{c}\varepsilon_{c},\;\sigma_{s}\;=\;E_{s}\varepsilon_{s}$$
(2)$$x_{0}/\varepsilon_{c}\;=\;h_{0}/(\varepsilon_{c}\;+\;\varepsilon_{s})$$
(3)$$\alpha_{1}E_{c}\varepsilon_{c}b\beta x_{0}\;=\;E_{s}\varepsilon_{s}A_{S}$$
(4)$$M\;=\;\alpha_{1}E_{c}\varepsilon_{c}b\beta x_{0}(h_{0}\;-\;\beta x_{0}/2)$$
where *σ_c_* and *σ_s_* were the sectional concrete maximum compressive stress and the tensile stress of the reinforcement bar. *E_c_* and *E_s_* were the elastic moduli of RUC and reinforcement bar, and *ɛ_c_* and *ɛ_s_* were the strain. The unmarked physical parameters mentioned above were formulated according to Standards [55]. When the bending moment was *M*, based on Equations (1)–(4), the stress level ($\bar{\sigma }$) was able to be calculated by
(5)$$\bar{\sigma }\;=\;\sigma_{c}/\sigma_{cu}\;=\;M/M_{u}$$
where *σ_cu_* was the concrete ultimate stress.

Well-reasoned design of the test setup was the basis for obtaining reliable results, especially for testing concrete dynamic properties [32]. The test setup and dynamic digital signal-processing instrument are shown in Figure 4.

Four INV9828 accelerometers were mounted on the top surface along the centerline of the beam, and the INV3062T Data Acquisition System recorded the acceleration-versus-time data. The version of this software was COINV DASP V10, and both the accelerometers and the software were developed 2014 by China Orient Institute of Noise & Vibration, Beijing, China. The sampling rate and signal acquisition time were 1024 Hz and three seconds, respectively, and the acceleration-versus-time data in three seconds were collected from the peak acceleration of 0.5 m/s^2^. Zheng et al. [42] reported that the concrete damping ratio increased with the increase of maximum response amplitude, and the damping property of rubberized concrete was more sensitive to the vibration response amplitude than that of plain concrete. This can be attributed to the increased Coulomb friction damping by adding the rubber particles [56,57]. Hence, the same excitation (300 N) of vibration was applied to the position of the concentrated load. The effect of excitation was slight, and considered to influence only the first model.

### 2.4. Theoretical Bases of Analysis

The dynamic analysis of concrete was based on structural dynamics [50].

(1) Natural frequency

The acceleration-versus-time data were obtained through the free vibration test of a cantilever beam under incremental loading. The test natural frequency of the specimens was determined with the attenuated curves of acceleration. For the concrete specimens without damage, their natural frequency can be calculated by
(6)$$\omega_{1}\;=\;2\pi f/\sqrt{1\;-\;\xi^{2}}\;=\;1.875^{2}\sqrt{EI/\left(\bar{m}L^{4}\right)}$$
where *ξ*, *E*, *I,* and *f* are the damping ratio, dynamic elastic modulus, the moment of inertial, and natural frequency, $1\;-\;\xi^{2}\;\approx \;1$. Also, $\bar{m}$ and *L* are the mass per unit length and calculated length of cantilever beams. Based on Equation (6), the theoretical natural frequency of the specimens was determined by
(7)$$f\;=\;1.875^{2}\sqrt{EI/\left(\bar{m}L^{4}\right)}/2\pi$$

(2) Damping ratio

The acceleration amplitude from the first 20 oscillation cycles was used to calculate *ξ* by
(8)$$\xi \;=\;\frac{1}{40\pi }\ln\frac{a_{1}}{a_{21}}$$
where *a*_1_ and *a*_21_ were the initial peak and the 21st peak in the given time history.

(3) Flexural dynamic stiffness

The properties of concrete change with the increase of rubber content and cumulative damage, and *EI* is no longer an invariant constant. Based on Equation (6), once the natural frequency was obtained, Equation (9) could be used to approximately predict the flexural dynamic stiffness under various conditions, and the dimensionless flexural dynamic stiffness was calculated by Equation (10).
(9)$$EI\;=\;{\left(2\pi f\right)}^{2}\bar{m}L^{4}/1.875^{4}$$
(10)$${\bar{\left(EI\right)}}_{i}\;=\;{\left(EI\right)}_{i}/{\left(EI\right)}_{0}\;=\;f_{i}^{2}/f_{0}^{2}$$
where (*EI*)_0_ and *f*_0_ were the flexural dynamic stiffness and natural frequency about cantilever beams of reference, and (*EI*)*_i_* and *f_i_* were the values of cantilever beams under different conditions. For each test of natural frequency, damping ratio and flexural dynamic stiffness, the average value of the two specimens was reported as the test value.

## 3. Results and Discussion

### 3.1. Slump

The concrete workability was measured by the slumps obtained by the standard test. The slumps of RUC and PRC with the same rubber content were similar, as listed in Table 2. The concrete slumps first decreased with the increase of rubber content, and then increased slightly when the rubber content exceeded 10%. The value reached the minimum when the content of the rubber was about 10%, and the slump reduced by 8.3% compared with that of NC. The slump of PRC4 reached the value of NC. All the rubberized fresh concrete mixtures exhibited a similar slump to that of NC.

In this test, the concrete slumps first decreased and then increased slightly with the increase of rubber content. This is understandable since both the amount of sand and water are the important factors of fresh concrete slumps. When some sand was replaced by rubber particles, the flowability of concrete decreased as the result of the reduction of sand and coarse aggregate ratio. Note that when the sand was replaced with enough rubber particles, the actual moisture content of RUC was higher due to the lower water absorption of rubber particles (1–2 mm) than that of sand, which may have increased the slump of concrete slightly. Also, the gas carried by rubber particles during the concrete mixing process can also improve the workability of the concrete.

### 3.2. Mechanical Properties

#### 3.2.1. Strength and Static Elastic Modulus

The test results for the concrete strength and static elastic modulus are listed in Table 3.

As most papers reported, the addition of rubber particles weakened the concrete strength significantly [28,56,57,58]. In this test, compared with NC, for the concrete with 10–20% rubber content, the cube and axial compressive strength reduced by about 15.0–42.0%, and static elastic modulus reduced by about 25.0–45.0%. To PRC with 5%, 10%, and 15% rubber content, their cube compressive strengths reached 1.3%, 7.1%, and 14.8% below that of NC, and their static elastic modulus were 13.5%, 23.2%, and 31.8% below that of NC. The concrete with added pretreated rubber particles showed a significant recovery of strength and satisfactory deformation. Pelisser et al. [46] reported that the compressive strength of PRC with 10% rubber content reduced by 14% (28 days) compared with NC, which is similar to the test result of this paper.

The effects of rubber content and pretreatment of rubber particles on the ratio of splitting tensile and axial compressive strength, and the ratio of flexural and axial compressive strength, are shown in Figure 5, where *f*_f_, *f*_cp_ and *f*_ts_ represent the dimensionless values of flexural strength, axial compressive strength, and splitting tensile strength. As shown in Figure 5, *f*_f_/*f*_cp_ is larger than *f*_ts_/*f*_cp_, and both the *f*_f_/*f*_cp_ and *f*_ts_/*f*_cp_ increase first and then decrease with the increase of rubber content. To *f*_f_/*f*_cp_ and *f*_ts_/*f*_cp_ of PRC, the values are larger than that of RUC. This indicated that the pretreatment of rubber particles with sodium hydroxide further enhanced the ductility of the rubberized concrete, and the recommended content of rubber was about 15%.

The surface of the routine and pretreated rubber particles, as well as their bonding interface with cement matrix, were obtained by scanning electron microscopy. Figure 6a–e show the surface of the unpretreated rubber particle, the surface of the pretreated rubber particle, the bonding interface between unpretreated rubber particle and cement matrix, the bonding interface of pretreated rubber particle and cement matrix, and the distribution of the rubber particles in concrete, respectively. The rubber particles are uniformly distributed in the concrete.

As shown in Figure 6, the surface of the pretreated rubber particle is clean and rough, and the width of an obvious crack along the interface between rubber and cement matrix reduces significantly after pretreatment. This can be attributed to the enhanced superficial adsorption, roughness, and the attenuation of hydrophobicity of rubber powders due to the corrosion of sodium hydroxide [28,46]. Therefore, the pretreatment of rubber particles helped reduce the internal defects between the rubbers and cement matrix, and increased the strength of RUC. However, the increase of concrete strength by enhancing the bonding interface between the rubber and cement matrix is not unlimited, since the rubber particles fill the voids in concrete as rubber aggregate, but they can’t act as the skeleton like sand due to its lower strength and stiffness.

Gao et al. [44] and Bompa et al. [56] reported that the added rubber particles increased the ductility of concrete, but concrete strength decreased significantly due to the rubber’s low stiffness and surface bonding with cement matrix. Hence, the ductility of the rubberized concrete may be further enhanced due to the elasticity of rubber and the enhanced interface between the rubber and cement matrix by pretreating the rubber particles. Moreover, lots of concrete internal defects due to a large content of rubber particles may reduce the splitting tensile and flexural strength more rapidly, which is adverse to concrete ductility, hence the values of *f*_f_/*f*_cp_ and *f*_ts_/*f*_cp_ decrease when the content of rubber particles exceeds 15%.

#### 3.2.2. The Ratio of Cracking and Ultimate Moment

The ratio of cracking and ultimate moment (*M_c_*/*M_u_*) of the cantilever beams reflects the ductility of the specimens. When *M_c_*/*M_u_* is too high, the material lacks ductility and there will be a sudden brittle failure of the specimen without obvious warning signs. The test results of *M_c_* and *M_u_* are listed in Table 4.

The effects of the content of rubber particles and the pretreatment on *M_c_*/*M_u_* are shown in Figure 7.

*M_c_*/*M_u_* of the specimens decreased with the increase of rubber content. Xie et al. [7] reported that the addition of rubber particles generally reduced the cracking load of the concrete specimens, but had a slight effect on the load-bearing capacity when the rubber content was less than 10%, which was similar to that of this research. This indicated that the addition of rubber particles increased the ductility of concrete significantly. As listed in Table 4, both *M_c_* and *M_u_* increase by pretreating the rubber particles, and the increase of *M_c_* is more significant than that of *M_u_*. This indicated that the addition of the appropriate amount of pretreated rubber particles increased concrete ductility, as well as the cracking property.

The uniformly distributed rubber particles in the cement matrix can effectively prevent the extension and expansion of micro cracks in the initial stage of concrete failure. However, it is completely possible that micro cracks will continue to extend along the cracks between rubber particles and cement matrix due to the unbonded interfaces and micro cracks between the rubber particles and cement matrix. The enhanced bonded interfaces between rubber particles and cement matrix are more beneficial for rubber particles bringing the advantage of elasticity and deformation into full play to effectively prevent micro cracks from extending and expanding, and to increase the ductility and crack resistance of concrete. It is also notable that the falling rates of *M_c_*/*M_u_* gradually decrease, and the decrease of *M_c_*/*M_u_* was not distinct when the rubber content exceeded 15%. This can be attributed to lots of concrete internal defects due to the large content of rubber particles. This indicated that the pretreatment of rubber particles with sodium hydroxide further enhanced *f*_f_/*f*_cp_ and *f*_ts_/*f*_cp_, and the recommended content of rubber is about 15%. Also, the compressive strength of PRC with 15% rubber content reached 14.8% below that of NC. Therefore, taking into account the strength, deformation, ductility, and crack resistance of concrete, the satisfactory content of pretreated rubber particles is about 15%.

### 3.3. Dynamic Properties

#### 3.3.1. Dynamic Elastic Modulus

The dynamic elastic modulus (*E_d1_*) about all the mixtures are listed in Table 3. Compared with NC, when the rubber content was 5%, 10%, 15%, and 20%, the reduction of dynamic elastic modulus was 3.8%, 5.8%, 10.5%, and 17.2% for RUC and 1.5%, 4.3%, 9.0%, and 14.9% for PRC, respectively. As for static elastic modulus, the pretreatment of rubber particles had a slight influence on dynamic elastic modulus.

As shown in Figure 8, with the same mix proportion, the dynamic elastic modulus is larger than the static elastic modulus, and the reduction of dynamic elastic modulus is smaller than that of the static elastic modulus. This is understandable since the dynamic elastic modulus is determined through a non-destructive test [42,43,53], and the effects of rubber particles on the concrete internal structure in the absence of external forces is much slighter than that in the condition of external load.

The dynamic elastic modulus represents the behavior of concrete deformation under dynamic load. The deflection of the specimen will decrease as the result of the increase of concrete dynamic elastic modulus. In this test, to PRC with 10% and 20% rubber content, their dynamic elastic modulus decreased by 4.3% and 14.9%, while the static elastic modulus decreased about by 23.2% and 45%. To RUC and PRC with 20% rubber content, the dynamic elastic modulus was higher than the static elastic modulus by 50.3% and 43.9%, and the difference was 50.0%, reported by Zheng et al. [42]. This indicated that the response of the specimen produced with PRC was smaller than that of the specimen produced with NC under the same dynamic load, and the anti-vibration and deformation of PRC were more satisfactory than that of NC.

#### 3.3.2. Natural Frequency

The acceleration-versus-time data in three seconds were collected from the peak acceleration of 0.5 m/s^2^ through the free vibration tests. The curves were typical logarithmic attenuated curves, and the partial curves are shown in Figure 9.

The test values of natural frequency were obtained with the curves, as listed in Table 4. The theoretical natural frequency of the specimens can also be determined by calculating with Equation (7), and the values are listed in Table 4. Small differences (less than 10.0%) between the test and theoretical values verified the validity of the test results.

As per the test natural frequency listed in Table 4, the increase of rubber content leads to a slight reduction, but an obvious decrease is observed with the increase of stress level, as shown in Figure 10.

The test natural frequency of the cantilever beams produced with NC (NCB) was 26.684 Hz, while the value reduced 2.8% and 2.5% for the specimens produced with RUC (RUCB) and PRC (PRCB), respectively, with 20% rubber content. To the simply supported beams in the similar test condition reported by Zheng et al. [42], the reduction of the natural frequency was almost the same with that in this test. Compared with NCB, under the stress level of 0.8, the natural frequency reduced 23.7% for NCB and 25–27.0% for RUCB and PRCB.

The natural frequency is a major dynamic parameter of concrete, which is mainly affected by dynamic elastic modulus and unit weight of the specimen. With the increase of rubber content, the natural frequency of concrete specimens decreases slightly since the reduction of dynamic elastic modulus is smaller than that of unit weight. The reduction of dynamic elastic modulus is the representation of development in concrete damage [53,59]. The natural frequency of concrete specimens decreases notably with the increase of stress level, which can be attributed to the increasing reduction of dynamic elastic modulus with the increase of damage accumulation (stress level).

#### 3.3.3. Flexural Dynamic Stiffness

The concrete dynamic elastic modulus (*E_d2_*) can also be determined through the free vibration test of cantilever beams, as listed in Table 3. There were some differences between *E_d1_* and *E_d2_*, but all of the differences are less than 11.2%, and less than 10.0% reported by Zheng et al. [42]. Therefore, it was an effective way to obtain the dynamic elastic modulus of concrete with some damage, and the flexural dynamic stiffness of specimens were determined by Equation (10). Compared with NCB, the flexural dynamic stiffness calculated with *E_d1_* and *E_d2_* of PRCB with 20% rubber content reduced by less than 15.0% and 11.6%. The flexural dynamic stiffness calculated with *E_d2_* reduced by 41.8% for NCB with the stress level of 0.8 and about 43.1–47.2% for RUCB (PRCB).

The effects of stress level on the damage (degeneration of flexural dynamic stiffness, $DI\;=\;1\;-\;\bar{EI}$) of these specimens are shown in Figure 11. Notably, the values of DI were approximate at the same stress level when the rubber content changed from 5% to 20%. This is understandable since the stress level has already included the influential factor of rubber content on the stiffness. Generally, the increase of stress level will lead to further concrete damage. During the concrete failure process, the redistribution of microstructure and stress is happening for each micro unit of concrete, and the rigid element transforms into the plastic element gradually with the increase of cumulative damage [60]. Therefore, the specimen can bear greater strength, but the stiffness reduces significantly. Kaewunruen et al. [21], Najim and Hall [43], and Akono et al. [61] reported that there was a correlation between stiffness (dynamic elastic modulus) and the compressive strength of concrete. According to much research into concrete fatigue behavior and the performance of damaged concrete, the stress level was an important parameter of concrete damage [62,63]. As shown in Figure 11, the data are limited by the number of experimental groups, but the trend of DI with the increase of stress level is similar to the typical curve of stiffness in the shape of an inverted “S”.

In this test, the relationship of DI and stress level can be expressed with a cubic equation, as shown in Figure 11. With this equation, the trend of DI with the increase of stress level can be predicted approximately, and the predicted point of inflection is 0.275 near the cracking stress level of concrete, which is as expected. This also confirms using the stress level to predict concrete damage related to the increase of the load.

#### 3.3.4. Damping Ratio

As the typical logarithmic attenuated curves of the acceleration obtained in this test show in Figure 9, the decay of acceleration increases with the increase of rubber content and stress level. The energy dissipation capacity of the specimens increased with the increase of rubber content and stress level. The damping ratio quantitatively expresses the energy dissipation capacity of concrete, and is determined based on these curves of the acceleration with Equation (8).

The pretreatment of rubber particles had little influence on the damping ratio in this test. The cantilever beams exhibited an obvious increase in damping ratio with the increase of rubber content and stress level. Compared with NCB, the damping ratio increased 10.6%, 24.5%, 44.4%, and 59.8% for RUCB and 6.3%, 19.5%, 40.5%, and 55.5% for PRCB when the rubber content was 5%, 10%, 15%, and 20%, respectively. In some reported references [26,42] under similar test conditions for undamaged specimens, the rise of damping ratio with the increase of rubber content was approximately the same. When the stress level was 0.6 and 0.8, the damping ratio increases by 60.0% and 179.7% for NCB, as well as 70–170% and 200–300% for RUCB and PRCB with 5–20% rubber content.

The addition of rubber particles will enhance the viscous energy dissipation capability of the cement matrix composite [39,40]. A large number of micro-interfaces were formed between rubber particles and the cement matrix because of the poor bonding between them [56,57], and the rubber particles distributed on the micro-interface of concrete increased Coulomb friction damping. Therefore, the damping ratio of PRC increased with the increase of rubber content, as shown in Figure 12.

In general, the higher the strength of concrete, the relatively worse are its deformation and dynamic properties [50,54]. The damping ratio of the specimen increases significantly with the increase of pretreated rubber content. It has been indicated that the compressive strength of PRC with 15% rubber content had reached 14.8% below that of NC. Therefore, the method of the superficial pretreatment of rubber particles provided in this paper can improve the strength of RUC to a certain extent, and ensure the original deformation and dynamic properties of RUC. In the field of current civil engineering where the requirement of strength is not too high, and the deformation and dynamic properties requirements are relatively strict, PRC has a good prospect of application.

The increase of damping ratio is slight in the elastic and elastic–plastic stage before the specimen cracking ($\bar{\sigma }\;<\;0.3$), since the small deformation can only dissipate a small amount of energy. After the appearance of cracks ($0.3\;\leq \;\bar{\sigma }\;<\;0.6$), the damping ratio increases evidently due to the friction of Coulomb. Then the damping ratio increases rapidly with the increase of cracks and the continuous development of cracks ($0.6\;\leq \;\bar{\sigma }\;<\;0.8$). The upward tendency of damping ratio during destruction is similar to the test result reported by Wang et al. [64]. The ratio of the cracking and ultimate moment (*M_c_*/*M_u_*) of the specimen decreases with the increase of rubber content, and the concrete failure period is prolonged. Therefore, the specimen will dissipate more energy with the increase of rubber content before its failure.

Recently, the measurement and calculation of the modal damping ratio of structures were advisable, and the damping ratio of the reinforced concrete structure was a function of displacement [50,65,66,67]. These methods reference measuring and calculating the damping ratio of concrete material [42,64,68,69].

However, the analysis of the damping ratio of RUC and PRC with the displacement during the destruction has some limitations, since the effect of rubber content on the displacement of the concrete specimens is significant. To the concrete specimens with different rubber content, with the same displacement, their cumulative damage widely differs from each other.

In this test, DI represented the degeneration of flexural dynamic stiffness and quantitatively expressed the degree of concrete damage. For all the specimens in this test, their values of DI have been obtained through the free vibration test. In order to show the effects of the damage and rubber content on the damping ratio of the specimens more intuitively, the three-dimensional Figure 12 was provided in this paper. The damping ratio showed linear growth with the increase of DI and rubber content, respectively. The relationship among *ξ*, DI and *ρ* can be expressed with a mathematical equation approximatively, as showed in Figure 12. Note that the equation is of great significance for reference in evaluating the damping ratio of rubberized concrete with some damage through the method of this paper.

## 4. Conclusions

The dynamic properties of PRC during the destruction were studied through a free vibration test with small cantilever beams under incremental loading in this paper. Also, a brief research of the static mechanical properties of PRC was also made. Some conclusions were obtained as follows:

The pretreatment of rubber particles with sodium hydroxide enhanced the strength, the ductility, and crack resistance of the rubberized concrete, and the optimum content of pretreated rubber particles was about 15%.With the same mix proportion, the concrete dynamic elastic modulus was larger than the static elastic modulus, and the reduction of dynamic elastic modulus was smaller than that of static elastic modulus with the increase of rubber content. Both the anti-vibration and deformation of PRC were more satisfactory than that of NC.The natural frequency and flexural dynamic stiffness of PRCB decreased significantly with the increase of cumulative damage, and the stiffness curve was in the shape of an inverted “S” with a point of inflection near the cracking stress level.Both the rubber content and the cumulative damage affected the concrete damping ratio significantly, and the damping ratio showed linear growth with the increase of DI (less than 0.8) and rubber content, respectively. PRC is a good choice where the requirement of strength is not too high, and the deformation as well as dynamic properties are relatively strict.

The research of dynamic properties presented in this paper is of great significant for reference to evaluate the dynamic properties of rubberized concrete with some damage. Moreover, considering the limitation of the test data, improvement of the test setup and specimens, to verify further applicability and validity of the method to measure the concrete dynamic properties, the extended numerical tests are the focus of our next work.

## Figures and Tables

**Figure 1 materials-14-02183-f001:**
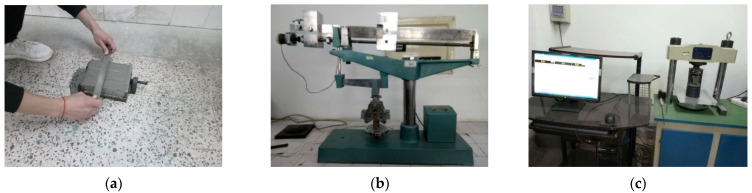
Preparation of cement mortar specimens and the test of their compressive and flexural strength: (**a**) Preparation of cement mortar specimens; (**b**) The test of cement mortar compressive strength; and (**c**) The test of cement mortar flexural strength.

**Figure 2 materials-14-02183-f002:**
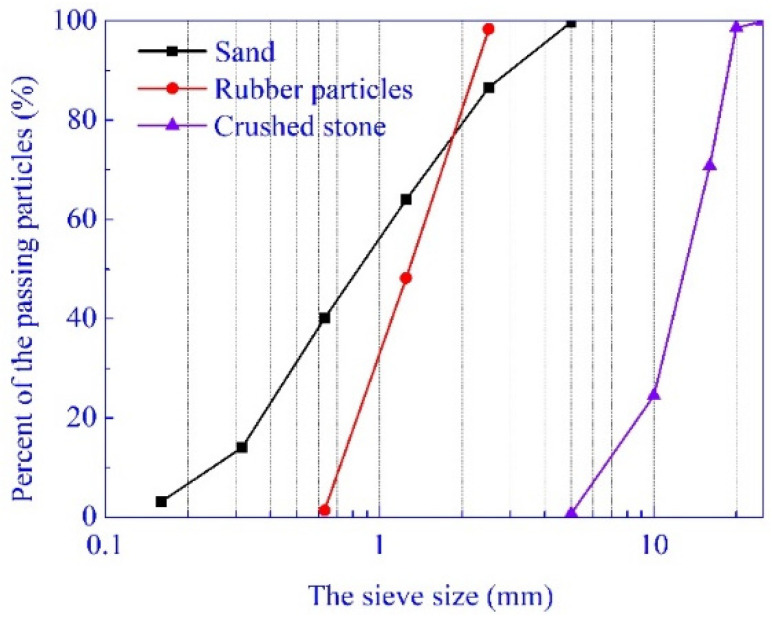
Gradation curves of the sand, rubber particles, and crushed stone.

**Figure 3 materials-14-02183-f003:**
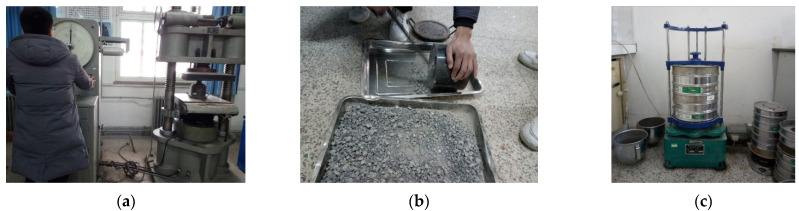
The test of coarse aggregate crushed index and the distributions of the rubber and aggregate particle size: (**a**,**b**) The test of coarse aggregate crushed index; (**c**) The test of distributions of the rubber and aggregate particle size.

**Figure 4 materials-14-02183-f004:**
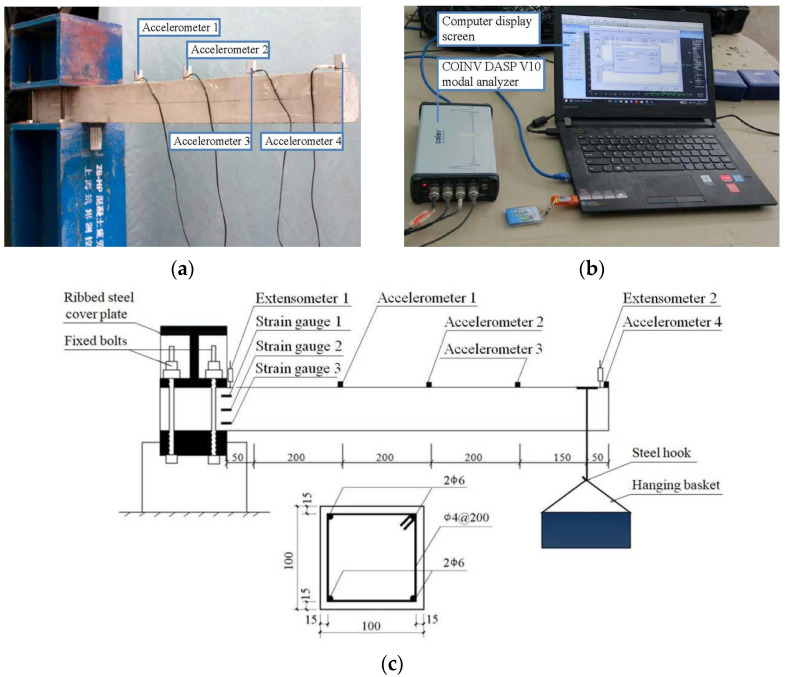
Test setup of free vibration and dynamic digital signal processing instrument: (**a**) Test setup of the cantilever beam; (**b**) Dynamic digital signal processing instrument; and ©(**c**) Details of the test setup.

**Figure 5 materials-14-02183-f005:**
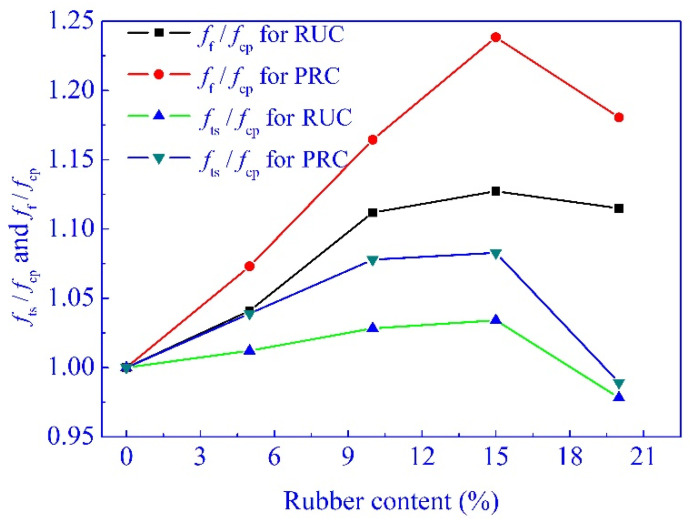
*f*_ts_/*f*_cp_ and *f*_f_/*f*_cp_ of rubberized concrete.

**Figure 6 materials-14-02183-f006:**
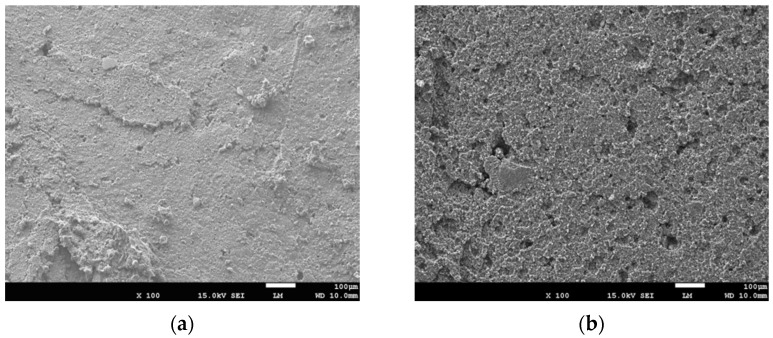
The surface of the rubber particles and their bonding interface with cement matrix: (**a**) The surface of routine rubber particles; (**b**) The surface of pretreated rubber particles; (**c**) The bonding interface of the rubber and cement matrix (×100); (**d**) The bonding interface of the pretreated rubber and cement matrix (×250); and (**e**) Distribution of the rubber particles in concrete.

**Figure 7 materials-14-02183-f007:**
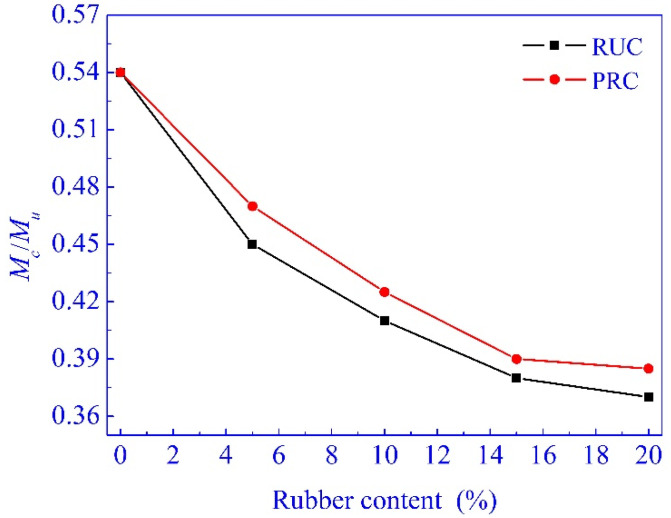
*M_c_*/*M_u_* of the specimens.

**Figure 8 materials-14-02183-f008:**
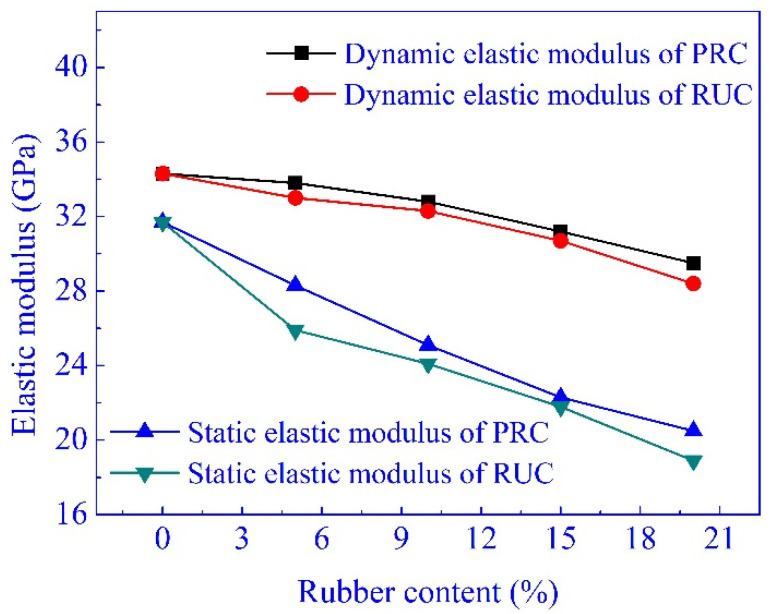
The dynamic and static elastic modulus.

**Figure 9 materials-14-02183-f009:**
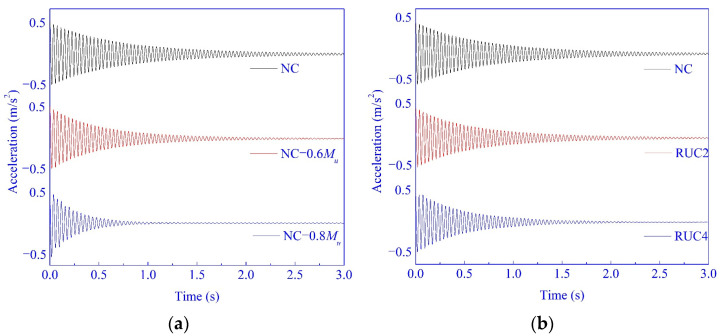
Time history of acceleration (with the data collected by Accelerometer 3): (**a**) The effects of rubber particle content; and (**b**) The effects of the cumulative damage.

**Figure 10 materials-14-02183-f010:**
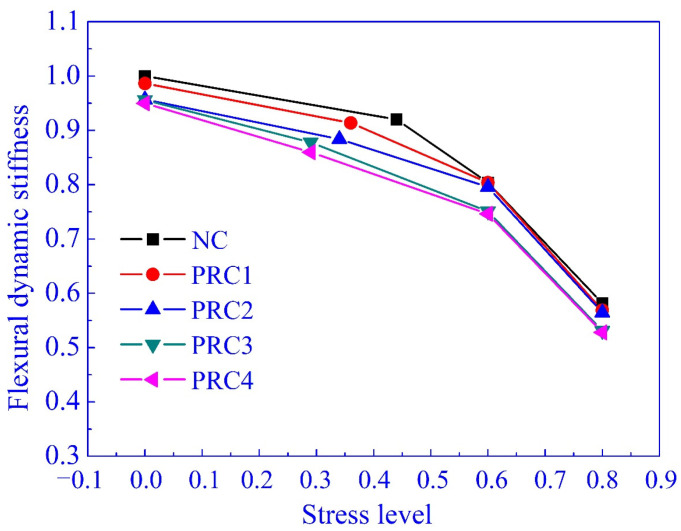
The effect of stress level on the natural frequency.

**Figure 11 materials-14-02183-f011:**
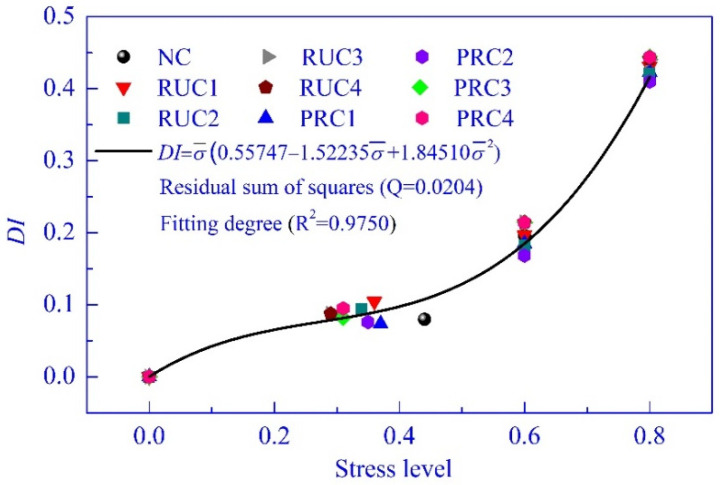
The test values of DI and prediction equation.

**Figure 12 materials-14-02183-f012:**
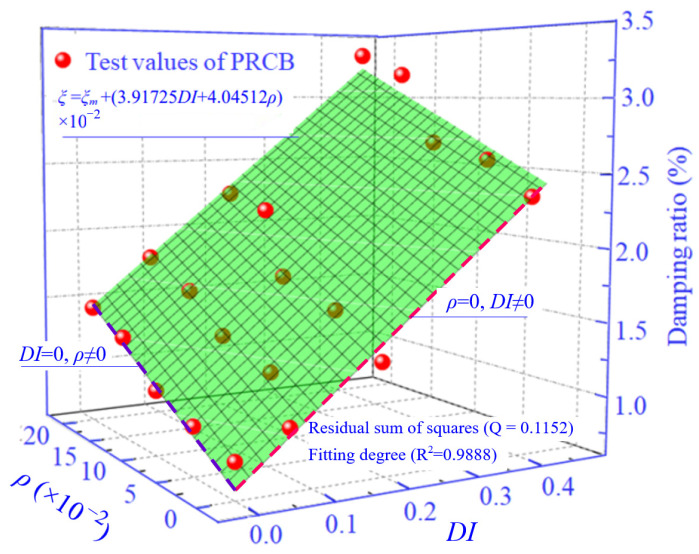
The test values of the damping ratio and prediction equation.

**Table 1 materials-14-02183-t001:** The properties of the cement, stone and sand.

**The Chemical Compositions of the Cement (%)**								
CaO		SiO_2_	Al_2_O_3_	Fe_2_O_3_	MgO	SO_3_	K_2_O	Na_2_O
61.63		21.00	5.20	3.92	2.75	2.48	0.83	0.20
**Physical and Mechanical Properties of the Cement**								
Setting time (min)		Compressive strength (MPa)		Flexural strength (MPa)		Density (kg/m^3^)	Blaine fineness (m^2^/kg)	Loss on ignition (%)
Initial	Final	3 days	28 days	3 days	28 days			
180	250	25.7	49.6	4.8	7.9	3100	348.3	1.05
**Properties of Crushed Stone and Sand**								
Aggregates		Size (mm)	Fineness	Apparent density (kg/m^3^)	Pile-up density (kg/m^3^)	Mud content (%)	Water Absorption (%)	Crushed index (%)
Coarse		5–20	--	2700	1628	0.47	1.00	14.8
Fine		0–4.75	2.7	2584	1520	1.80	1.30	--

**Table 2 materials-14-02183-t002:** The mix proportions and slumps of concrete.

Mixtures	Rubber Content	Mix Proportions (kg/m^3^)						Slump (mm)
		Cement	Water	Unpretreated Rubber	Pretreated Rubber	Crushed Stones	Sand	
NC	0	325.0	195.0	0	0	1092.0	728.0	60
RUC1	5%	325.0	195.0	17.61	0	1092.0	691.6	57
PRC1	5%	325.0	195.0	0	17.61	1092.0	691.6	60
RUC2	10%	325.0	195.0	35.22	0	1092.0	655.2	55
PRC2	10%	325.0	195.0	0	35.22	1092.0	655.2	50
RUC3	15%	325.0	195.0	52.83	0	1092.0	618.8	57
PRC3	15%	325.0	195.0	0	52.83	1092.0	618.8	54
RUC4	20%	325.0	195.0	70.44	0	1092.0	582.4	61
PRC4	20%	325.0	195.0	0	70.44	1092.0	582.4	65

**Table 3 materials-14-02183-t003:** The strengths and elastic modulus of concrete.

Mixtures	Strength (MPa)				Static Elastic Modulus (GPa)	Dynamic Elastic Modulus (GPa)		
	Cube Compressive	Axial Compressive	Flexural	Splitting Tensile		*E_d1_*	*E_d2_*	Difference (%)
NC	37.8	25.8	5.60	2.69	31.7	34.3	30.9	9.9
RUC1	36.2	23.5	5.31	2.48	25.9	33.0	29.6	10.3
RUC2	33.3	20.8	5.02	2.23	24.1	32.3	28.9	10.5
RUC3	29.0	17.9	4.38	1.93	21.8	30.7	27.3	11.1
RUC4	23.4	15.0	3.63	1.53	18.9	28.4	26.8	5.6
PRC1	37.3	24.0	5.59	2.60	28.3	33.8	30.0	11.2
PRC2	35.1	21.8	5.51	2.45	25.1	32.8	30.7	6.4
PRC3	32.2	18.6	5.00	2.10	22.3	31.2	28.4	9.0
PRC4	27.4	16.0	4.10	1.65	20.5	29.5	27.3	7.5

**Table 4 materials-14-02183-t004:** Cracking moment, ultimate moment, and natural frequency of cantilever beams.

Mixtures	*M_c_*	*M_u_*	Frequency (Hz)	
			Theoretical	Test
NC	1.42	2.62	30.18	26.68
RUC1/PRC1	1.13/1.17	2.55/2.53	29.01/29.36	26.51/26.50
RUC2/PRC2	1.03/1.07	2.43/2.44	28.70/28.92	26.26/26.10
RUC3/PRC3	0.87/0.91	2.30/2.32	27.98/28.21	26.01/26.09
RUC4/PRC4	0.80/0.89	2.18/2.22	26.91/27.43	25.92/26.01

## Data Availability

Data is contained within the article.

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
