# Peer review of "Dynamic Properties of Pretreated Rubberized Concrete under Incremental Loading"

_materials, 2021, doi:10.3390/ma14092183_

Round 1
Reviewer 1 Report
The current study investigates the dynamic performance of pre-treated concrete with rubber like properties by studying the flexural dynamic stiffness and damping ratio of the material. The authors reported that pre-treatment of the rubber particles helped improve strength, ductility, and crack resistance performance. It also enhanced energy dissipation for the concrete material when being destructed.
The abstract needs some restructuring and rephrasing. Please consider reviewing the abstract and highlight the novelty, major findings and conclusions. In simple and concise manner.
In the introduction, there is a lot of bulk citations which is not acceptable unless the authors give full credit to each reference mentioned. For example in 20 lines there is over 35 reference? This is not acceptable and must be fixed
Although the authors provide somewhat good introduction but it still lack critical literature review of past studies, report on what they have done and what were their main findings, then mention how does this study brings new knowledge and difference to the field.
Also please answer the following question in the introduction: What is the research gap did you find from the previous researchers in your field? Mention it properly. It will improve the strength of the article.
Line 86-96 please combine in one larger paragraph
Please avoid writing small paragraphs of 2-3 lines and combine them together into bigger ones
In the materials section please add some images for the samples produced and the machines/equipment used for making them. This is an experimental study and must be clearly detailed in how the experimental work was done.
Please combine tables 1-3 in one larger table
Table 1-3 please add references of the source of this information unless you measured it by yourself.
Table 4 why the authors used those specific ratios in their study, is it according to open literature or recommendation from industry or randomly chosen?
Please add list of nomenclature for all the symbols and Greek letters reported in this work at the start or end of the manuscript
Line 199 what do the authors mean by good workability, according to what application or standard..etc.
Line 200 “the concrete slumps first decreased..” what decreased and increased? Workability
The manuscript requires average to extensive English editing and spell check
Line 236-238 so this is an important conclusion, how about past studies, did they find something similar or different to your findings? Please discuss further and support with references
Line 242-243 “. Due to the elasticity of rubber and the enhanced int….” this claim needs a reference to support it
Figure 4 please add some text to tell us what exactly are we looking at here
Line 260 “hat the addition of rubber particles increased the ductility of concrete significantly” please support this claim with references and discuss further
overall good study and results
Reviewer 2 Report
Dear Authors,
I read your contribution with interest and attention. I can surely say that there are several remarkable points for reflection and, in general, that the manuscript is valid. At the same time, I have to say that there are several issues that require correction or explanation.
Kind regards
- Authors should be commended for thorough and well-thought-out literature review.
- The novelty of the work should be clearly indicated in the Introduction.
- Please provide the description of the manuscript structure at the end of the introductory part.
- Introductory sentence would be advisable at the beginning of Section 2.1., I think that adding it would be better that starting Section 2.1. with: “Cement: The […]”
- Authors should be commended for a thorough description of materials used and samples preparation.
- Line 121: Please indicate how the calculations for determining natural frequencies were conducted.
- Line 136-138: what was the incremental loading imposed on the structure?
- Line 152: Please provide the detailed description of the test setup and indicate the signal accusation parameters used (see 3390/ma13071630)
- Line 156-160: How the differences in maximum response amplitude affected the damping ratio, please explain.
- Figure 7: data from which accelerometer was used to plot graphs.
- Line 317: how the natural frequency value was obtained.
- Line 173: why “, was added, please comment
Reviewer 3 Report
In the paper are presented a series of information on dynamic properties of pretreated rubberized concrete under incremental loading
From the analysis of the information presented in the article, I found the following:
- The paper presents a series of results that could be of interest to the scientific community:
- The Materials section must be substantially developed, in the sense that the properties of the rubber particles must be presented. This is necessary because rubber is the material that, depending on the added quantities, can influence the properties of cement structures. It is also necessary to correlate the information on the quantities of rubber particles presented in line 104 with those presented in Table 4. To this end, the same unit of measurement should be kept;
- Within the research methodology, the decision to use a maximum value of 20% rubber particles in the structure of the samples must be justified;
- How to test the samples using incremental loads must be detailed;
- It is necessary to present macroscopic images of the samples made in order to be able to observe the distribution of rubber particles in their structure;
- Only part of relations 1-10 are used in the section presenting the experimental results and thus it is necessary that those relations that were not used be eliminated;
- The abbreviations NCB and PRCB should be specified more clearly;
- The discussion part needs to be improved in order to better highlight the novelty brought by the research presented in the paper compared to other research in the field. In the current form in the discussion part, references are made to a series of bibliographic sources through which the obtained results are confirmed, but the novelty brought by the research cannot be identified;
- In the final part of the conclusions, the future research directions must be presented. The practical applications of the research could also be presented in conclusions.
Round 2
Reviewer 1 Report
all questions answered
Reviewer 2 Report
I can now reccomend manuscript for publication
Reviewer 3 Report
The authors revised their manuscript according to my suggestions. Thus the manuscript can be accepted for publication.